# Entropy by Neighbor Distance as a New Measure for Characterizing Spatiotemporal Orders in Microscopic Collective Systems

**DOI:** 10.3390/mi14081503

**Published:** 2023-07-26

**Authors:** Yulei Fu, Zongyuan Wu, Sirui Zhan, Jiacheng Yang, Gaurav Gardi, Vimal Kishore, Paolo Malgaretti, Wendong Wang

**Affiliations:** 1University of Michigan—Shanghai Jiao Tong University Joint Institute, Shanghai Jiao Tong University, Shanghai 200240, China; 2Department of Mechanical Engineering, Carnegie Mellon University, 5000 Forbes Ave, Pittsburgh, PA 15213, USA; 3College of Engineering, University of Michigan, Ann Arbor, MI 48109, USA; 4The Academy for Engineering and Technology, Fudan University, Shanghai 200433, China; 5Physical Intelligence Department, Max Planck Institute for Intelligent Systems, 70569 Stuttgart, Germany; 6Department of Physics, University of Stuttgart, 70569 Stuttgart, Germany; 7Department of Physics, Banaras Hindu University, Varanasi 221005, India; 8Helmholtz Institute Erlangen-Nürnberg for Renewable Energy (IEK-11), Forschungszentrum Jülich, 52425 Jülich, Germany

**Keywords:** active matter, collective behavior, phase transition, order parameter

## Abstract

Collective systems self-organize to form globally ordered spatiotemporal patterns. Finding appropriate measures to characterize the order in these patterns will contribute to our understanding of the principles of self-organization in all collective systems. Here we examine a new measure based on the entropy of the neighbor distance distributions in the characterization of collective patterns. We study three types of systems: a simulated self-propelled boid system, two active colloidal systems, and one centimeter-scale robotic swarm system. In all these systems, the new measure proves sensitive in revealing active phase transitions and in distinguishing steady states. We envision that the entropy by neighbor distance could be useful for characterizing biological swarms such as bird flocks and for designing robotic swarms.

## 1. Introduction

Active matter systems consisting of self-propelled or driven individuals can self-organize and form large-scale ordered spatiotemporal structures [1,2]. They serve as model systems for non-equilibrium mechanics as their individual units break the time-reversal symmetry [3] and local detailed balance [4]. They are ubiquitous across many scales, from bird flocks [5], fish schools [6], and ant colonies [7] to bacteria colonies [8,9], artificial micro-nano machines [10,11,12,13,14,15,16,17], and molecular motors [18,19]. Though these collective systems differ widely in the sizes and levels of intelligence of their individual units, they share the common characteristics of self-organization, i.e., local rules and behaviors determining global properties and functions. Therefore, one of the fundamental challenges in the study of collective systems is constructing appropriate measures based on local parameters that reflect global properties or functions. Addressing this challenge will contribute to our understanding of the principles of self-organization in all collective systems and is meaningful for both understanding the living systems and designing artificial systems.

Collective systems are often analyzed from the perspective of the statistical mechanics of particles. In this approach, the change in the spatiotemporal patterns of a collective system can be characterized by an order parameter, akin to the characterization of phase transition in equilibrium systems. In the seminal work of Vicsek, the average velocity was used as the order parameter to study the phase transition of collectives of self-propelled particles [20]. This average velocity is the vectorial mean of individual self-propelled particles and thus reflects the degree of alignment of the whole collective. This average velocity, however, does not reflect the degree of order based on the positions of individual units. Positional orders of static structures are mostly analyzed according to the symmetries of point groups, a typical example being crystal structures. Analyzing dynamic spatiotemporal patterns of collective systems poses challenges for this approach because the relative positions between individual units vary over time.

An alternative approach is to analyze the spatiotemporal patterns from the perspectives of information. This approach has a long history but is under resurgence recently. The connection between structure and information can be traced back to the idea of an “aperiodic crystal” in living systems by Erwin Schrödinger, which foretold the discovery of DNA [21]. Following the ideas of Kolmogorov–Chaitin (KC) complexity, proposed in the 1960s to quantify the amount of information that one computer program carries, Martiniani et al. recently proposed computable information density (CID) to measure the information contents contained in the structures of non-equilibrium systems [22]. Following the spirit of KC complexity, Mackay and Cartwright noted that structures also have repetition and redundancy, just the same as computer programs, and they proposed assembly complexity to measure the amount of information a structure carries [23]. Other researchers seek to apply the formalism of Shannon entropy to characterize structures. For example, Frenkel et al. proposed Voronoi entropy (VE) to quantify the symmetry and orderliness of two-dimensional microdroplet clusters and compared it with continuous symmetry measure (CSM) [24]. As another example, the Shannon entropy could be used to quantify the degree of order in crystallography and the complexity of the inorganic crystal [25,26]. In addition, information theory has been widely applied to crystallography and has given rise to chaotic crystallography [27]. Building on these ideas, we recently proposed the entropy by neighbor distance $H_{NDist}$ as a measure of the order of the collective pattern generated by micron-scale spinning rafts driven by magnetic fields [28]. This measure can distinguish various patterns of hundreds of units at different rotating frequencies of magnetic fields and is even sensitive to different local symmetries of individual rafts. As a natural extension, we want to ask the following question: How well does this new measure generalize to other collective systems?

Here, we extend the new measure entropy by neighbor distance $H_{\mathrm{NDist}}$ to three types of systems: a simulated self-propelled boid system, two active colloidal systems, and one centimeter-scale robotic swarm system. In the simulated boid system, we compare $H_{\mathrm{NDist}}$ with the average velocity $v_{a}$ in characterizing the change in spatiotemporal patterns. We compare the situation where neighbors are defined by metric pairwise distances with the situation where neighbors are defined by topology and find that $H_{\mathrm{NDist}}$ is more sensitive for the latter case. In active colloidal systems, we demonstrate that $H_{\mathrm{NDist}}$ correlates well with the system’s response to external stimuli and can also distinguish different steady-state patterns. We analyze the impact of bin size in the calculation of $H_{\mathrm{NDist}}$. In the centimeter-scale robotic swarm system, we show that $H_{\mathrm{NDist}}$ can track the overall progress of self-organization and distinguish different steady-state patterns. Moreover, the analysis of circular agents assembled into the configuration of the 26 letters in the alphabet shows the shape dependence of $H_{\mathrm{NDist}}$. We expect that these results will further stimulate efforts to apply $H_{\mathrm{NDist}}$ in other areas, particularly in biological systems and microrobotic systems.

## 2. Results and Discussion

### 2.1. Simulated Boid System

We adopt a discrete-agent model with point-like boids whose dynamics are governed by three rules: cohesion, separation, and alignment [29]. Cohesion and separation cause the attractive and repulsive interactions between neighboring boids, respectively, and the alignment rule causes a boid to align its heading direction with the average of its neighbors. The sum of these interactions is scaled by a coefficient *S*, the steering factor, and then added to the velocity of the boid in order to get the new velocity of the boid at the next time step. The steering factor *S* adjusts the relative weight between the influence of the neighbors and the inertial of the boid. We use two methods to define neighbors in our simulation. The first one is through metric distance: If two boids are within a certain threshold distance, they are neighbors. The second one is through topology: We construct Voronoi tessellations to determine neighboring pairs. The initial position and velocity direction of all boids are random (assuming uniform distribution throughout the entire interval), and the amplitude of the velocity is kept constant. At each step of the update on velocity, we introduce noise in the direction of velocity to investigate the effect on the collective patterns. A detailed description of the model of the simulation is included in the method section. Two representative videos are included in the Appendix A.

First, we use the result of a representative single simulation run to demonstrate the temporal evolution of the collective patterns (Figure 1). This run is performed with the condition of zero noise, and the neighbors are defined by topology. Figure 1a shows the patterns of boids at several key time steps. In the beginning, the positions and the directions of velocities of boids are random. As the simulation progresses, the directions of the boids are gradually aligned. Figure 1b,c show the quantification of these patterns of boids. Figure 1c shows that $v_{a}$ starts at zero in the initial state and continues to increase until reaching near the maximum value of 1 after ~40,000 steps, while the topological structures revealed by $H_{\mathrm{NDist}}$ do not appear to change much before 40,000 steps, and the value of $H_{\mathrm{NDist}}$ only starts to decrease after 40,000 steps, and then reaches a plateau after 80,000 steps (Figure 1b). This change in $H_{\mathrm{NDist}}$ is reflected by the increase in uniformity of the distribution of boids in the patterns. The comparison of Figure 1b,c shows that the velocity alignment is faster than the change in topology structures in the patterns of the boids model. It demonstrates that the order of the velocity direction and the topological structure are not equivalent, so these two measures characterize different aspects of the collective patterns.

Next, we explore the effect of noise in boid velocity updates and the effect of two different methods of defining neighbors on the characterization of collective patterns by $H_{\mathrm{NDist}}$ and $v_{a}$ (Figure 2). Noise is added to the angle component of the velocity update at each time step, $v_{i}\left(t+\Delta t\right)=V\hat{v}+\left(\theta_{i}+\Delta \theta \right)\hat{\theta }$, where *V* is the constant speed of all boids, and a random variable Δθ is added to the angular component of the velocity. Then, Δθ is drawn from a uniform distribution $\left[-\eta ,\eta \right]$, and η is the noise factor that is varied from 0 to 7°. We run the simulations with different noise factors and take the data of the last 1000 steps when the simulations reach steady states.

In the situation where the neighbors are defined by topology (Figure 2a), after the final steady-states are reached, the magnitude of the averaged velocity $v_{a}$ decreases as the noise factor η increases, whereas entropy by neighbor distance $H_{\mathrm{NDist}}\;$ increases as η increases. The onset of the change in $v_{a}$ and $H_{\mathrm{NDist}}$ also differ: $H_{\mathrm{NDist}}$ is sensitive to the initial small noise, whereas $v_{a}$ is robust to the small noise and only decreases appreciably when η reaches 3°~4°. Intuitively, noise influences the velocity direction directly, but the small noise does not greatly change the velocity alignment, so $v_{a}$ is relatively stable with the small noise. However, because $H_{\mathrm{NDist}}$ is based on the topological structure, a small fluctuation may cause the change of the topology. For example, the number of neighbors in a hexagonal lattice could change from 6 to 5 or 7 because of a small shift in the position of one vertex. This sensitivity explains why $H_{\mathrm{NDist}}$ increase dramatically at the beginning. Therefore, these two measures are again complementary: one for the topological order and the other for the alignment order.

In situations where neighbors are defined by metric distances (Figure 2b), $v_{a}$ still exhibits a significant decrease as the noise factor η increases, whereas $H_{\mathrm{NDist}}$ shows little changes as η increases. We suspect that the reason for the negligible change in $H_{\mathrm{NDist}}$ is that boids may interact with a lot more neighbors when the neighbors are defined by metric distances than when the neighbors are defined by topological structures, and hence they tend to form big clusters in the final steady state. This tendency of cluster formation is relatively robust to the disturbance of noise. Therefore, $H_{\mathrm{NDist}}$ shows negligible changes.

### 2.2. Active Colloids

Active colloidal systems include self-propelled microparticles [30,31,32] and external field-driven microparticles [33,34,35]. They not only serve as model systems for research on non-equilibrium physics [36,37,38] but also are used as microrobots for biomedical and environmental applications [39,40,41]. One of the striking phenomena in active self-propelled particles is that they could experience so-called motility-induced phase separation (MIPS) with purely repulsive interactions, which is impossible for passive colloidal particles without attraction. This phenomenon has been observed in both simulations and experiments [7,42,43].

One of the representative models for MIPS is the light-activated colloidal particles. For example, Palacci et al. used a suspension of synthetic photoactivated colloidal particles [44], and they observed that when the blue light is on, homogeneously distributed particles began to assemble into clusters with an average size of 35 particles (~10 μm). They claim that the osmotically driven motion and steric hindrances (collisions) are necessary for the formation of “living crystals”. We tracked the particles in a video of living crystals and performed analysis using $H_{\mathrm{NDist}}$ (Figure 3a and Appendix A). $H_{\mathrm{NDist}}$ captures the three stages of the whole process: clustering (light on), dispersed (light off), and again clustering (light on). Turning on the light causes the particles to move, which induces MIPS, making the structure more ordered and clustered, and thus causing $H_{\mathrm{NDist}}$ to decrease, and the reverse is true for the dispersed phase.

Further, we explore the influence of bin sizes on calculating $H_{\mathrm{NDist}}$. Figure 3b shows that the trends differ when the bin sizes are smaller or larger than the radius *R* of the particles (3 px). For example, at 23 s (light off) the $H_{\mathrm{NDist}}$ increases or decreases for the bin sizes smaller or larger than *R*, respectively. In addition, the downward trend that appeared in 34 s (light on) is not well reflected in the three curves whose bin sizes are larger than *R*. The analysis of $H_{\mathrm{NDist},\mathrm{norm}}$ also shows that the trend in the three curves whose bin sizes are smaller than *R* is similar (Figure 3c).

To explain the effect of the bin sizes on $H_{\mathrm{NDist}}$, we check the histograms. When the bin size is too large, small changes in separation distances are not accounted for, resulting in $H_{\mathrm{NDist}}$ being not sensitive enough, such as the comparison before and after 23 s (light off). Figure 3d shows that after 23 s, the distribution near the first peak increases when the bin size is 1 px, which contributes to the increase of $H_{\mathrm{NDist}}$, whereas it is difficult to capture these small changes in distributions when the bin size is 10 px. Furthermore, by comparing the histograms at the beginning and at the end, we find that the initial neighbor distance distribution is concentrated in the range below 100 px. If the bin size is too large, the corresponding $H_{\mathrm{NDist}}$ of the initial dispersed state will be smaller than the final aggregated state, which is inconsistent with our intuition. Therefore, we think that bin sizes smaller than the radius of particles are good choices for the analysis of collective microscopic patterns.

**Figure 3 micromachines-14-01503-f003:**
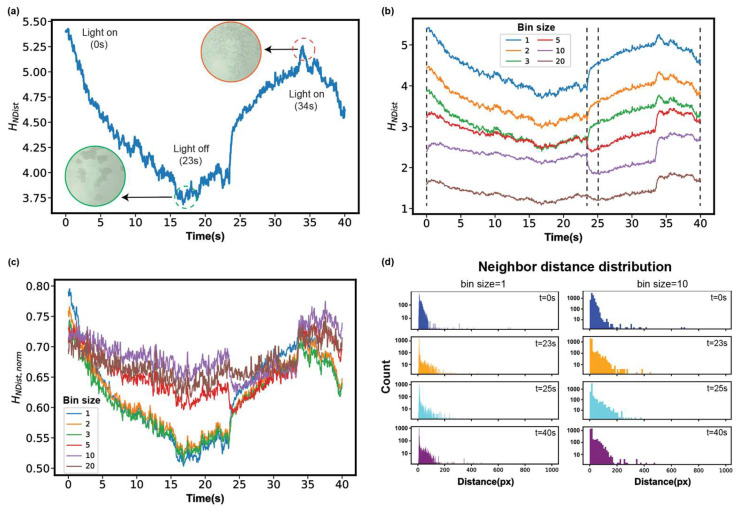
The analysis of light-activated colloidal surfers system using entropies by neighbor distances. (**a**) Temporal evolution of $H_{\mathrm{NDist}}$ (bin size = 1 px). (**b**) $H_{\mathrm{NDist}}$ of different bin sizes (px). (**c**) Temporal evolution of $H_{\mathrm{NDist},\mathrm{norm}}$ with different bin sizes (px). (**d**) Histogram of neighbor distance distribution at four critical moments with bin size 1 and 10 px. The original video is from the Ref. [44].

Another typical type of experimental model of active colloids is the electric-field-activated colloidal particles. Snezhko et al. used pear-shaped dielectric particles confined inside a cylindrical cell to introduce chiral rollers with activity-controlled curvatures of their trajectories and spontaneous handedness of their motion [45]. By controlling activity through variations of the static (DC) electric field applied perpendicular to the bottom surface of the cell, they showed emergent dynamic phases, ranging from a gas of spinners to aster-like vortices. We tracked the particles in two of their published videos and demonstrate that $H_{\mathrm{NDist}}$ could distinguish a gas-like steady state and an aggregated steady state easily **(**Figure 4a and Appendix A).

Furthermore, the two states remain distinguishable as long as the bin sizes are smaller than the radius of the particles (*R* = 10 px), which reflects the robustness of $H_{\mathrm{NDist},\mathrm{norm}}$ in distinguishing patterns with large differences in their orders (Figure 4b). In examining the non-normalized $H_{\mathrm{NDist}}$, we find that $H_{\mathrm{NDist}}$ decreases with increasing bin sizes (Figure 4c). Different bin sizes cause the overlapping of curves of the two states, and the fluctuation of non-normalized $H_{\mathrm{NDist}}$ of the gas state is greater than that of the aggregated state. Figure 4d shows that the standard deviation (std) of the gas state is nearly three times that of aggregated state. In addition, the values of standard deviations are stable under different bin sizes, which provide additional support for choosing the bin size smaller than *R*, as the previous case suggests.

**Figure 4 micromachines-14-01503-f004:**
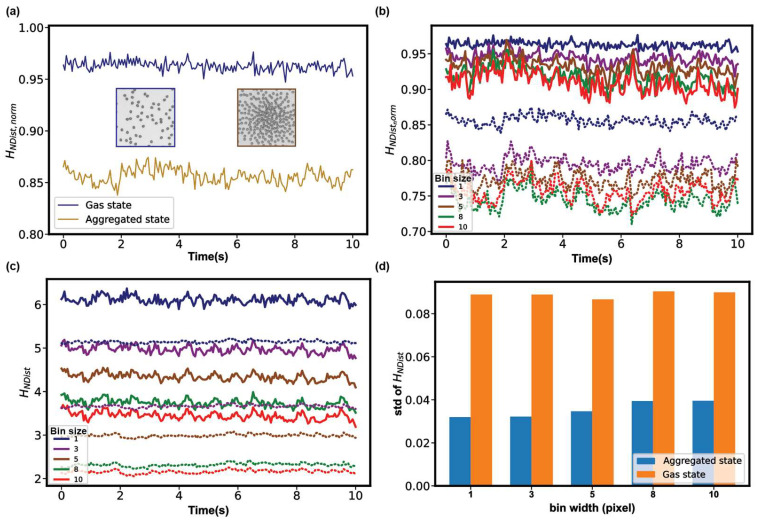
The analysis of active chiral fluids with two different states using entropies by neighbor distances. (**a**) Temporal evolution of $H_{\mathrm{NDist},\mathrm{norm}}$. (**b**) $H_{\mathrm{NDist},\mathrm{norm}}$ of different bin sizes (px): solid lines represent the gas state and dotted lines represent the aggregated state. (**c**) Non-normalized $H_{\mathrm{NDist}}$ of different bin sizes (px): solid lines represent the gas state and dotted lines represent the aggregated state. (**d**) The standard deviations (std) of $H_{\mathrm{NDist}}$ of the time series. The original videos are from the reference [45].

### 2.3. Robotic Swarm

Swarm robotics is the study of a large number of robots that can spontaneously determine their next move by exchanging information, such as their position and speed, with neighboring individuals, and self-organize into the desired configuration without sending and receiving direct control commands from external supervisors such as a base station or a human overseer [46]. This bionic intelligent control system takes inspiration from the self-organized behaviors of social animals and has important applications in many fields ranging from farming and food management, to military defense and space systems [47].

Nagpal et al. demonstrate a thousand-robot swarm (dubbed kilobots) capable of large-scale, flexible self-assembly of two-dimensional shapes entirely through programmable local interactions and local sensing [48]. Specifically, they developed a collective algorithm that enables 1024 robots, each with limited capabilities, to cooperatively assemble into some pre-designed shapes. We picked two of their published videos: the assembly of the letter K-shape and starfish shape, and we tracked the robots during the self-assembly process (Figure 5a,b and Appendix A). $H_{\mathrm{NDist}}$ show a continuous gradual increase in both self-assembly processes and thus can be a sensitive measure for the progress of the programmable self-assembly. Moreover, the final values of $H_{\mathrm{NDist}}$ for the two shapes differ, which suggests $H_{\mathrm{NDist}}$ may be used as an indicator for reconfigurable kilobot self-assemblies as well.

To further demonstrate that $H_{\mathrm{NDist}}$ is shape-dependent, we simulate robot swarms assembled into configurations of 26 letters in the alphabet. We first create a densely packed lattice composed of circles of equal diameter, and then obtain the configurations of letters by masking the lattices with corresponding shapes (Figure 5c). These letters show different entropy values (Figure 5d). These differences are due to the circles along the edges: Even though the configurations are all based on the square lattice, their edge profiles are quite different, and the units on edges can show significantly different neighbor distances, resulting in different neighbor distance distributions. This finding may be useful for the design of control algorithms for robotic swarms.

**Figure 5 micromachines-14-01503-f005:**
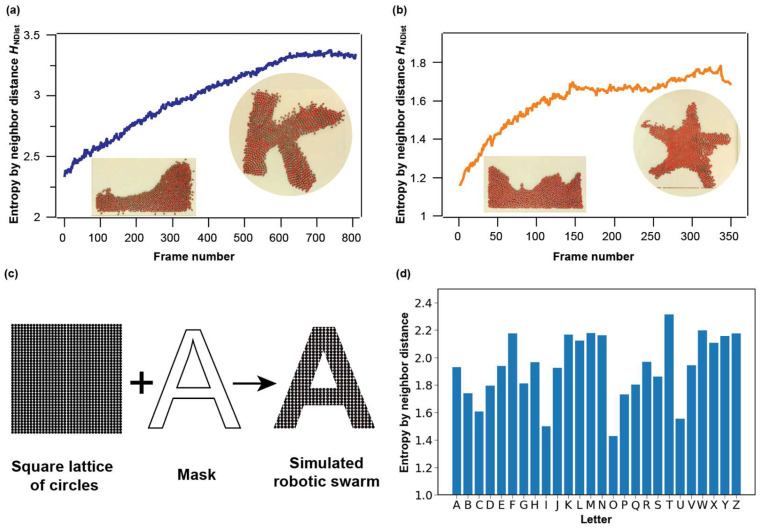
The analysis of the self-assembly process in the kilobot swarm using $H_{\mathrm{NDist}}$. (**a**) K-shape. (**b**) Starfish shape. The original videos are from the Ref. [48] (**c**) simulated robot swarm assembled into shapes of letters. (**d**) $H_{\mathrm{NDist}}$ of 26 letters in the alphabet.

## 3. Conclusions and Discussion

In summary, we extend $H_{\mathrm{NDist}}$ into three types of active systems and prove its usefulness in characterizing the spatiotemporal patterns of these systems. In the active colloid systems and robotic swarm systems, we have analyzed published videos of experiments and showed $H_{\mathrm{NDist}}$ as a useful indicator of pattern changes and a measure to distinguish different steady states. In the simulated boid system, we showed that $H_{\mathrm{NDist}}$ is a sensitive measure of phase transitions and can reveal the existence of noise. We further found that $H_{\mathrm{NDist}}$ is more sensitive to situations where neighbors are defined by the topological structures instead of metric distances. Field studies of starling flocks show that the interactions of birds depend on topological distance rather than metric distance, and each bird only interacts with six to seven neighbors on average [49]. Accordingly, we surmise that $H_{\mathrm{NDist}}$ could be a sensitive measure for biological systems, including bird flocks.

We think that the reason why $H_{\mathrm{NDist}}$ could accurately reveal the changes of patterns in a variety of systems relates to the dominance of pairwise contribution to the configuration entropy in a system. The configuration entropy of a system contains contributions from multi-body correlation functions, and Duane Wallace et al. have shown that in simple liquids the two-body excess entropy, which only involves the pair correlation function, accounts for more than 90% of the configuration entropy [50,51,52]. We are currently investigating the fundamental relation between $H_{\mathrm{NDist}}$ and two-body excess entropy, and the results will be published in the future.

We also anticipate useful applications of $H_{\mathrm{NDist}}$ in swarm robotics because a local measure such as $H_{\mathrm{NDist}}$ should in principle be very sensitive to the self-organization of robots who only interact locally. For example, in recent years researchers have been developing decentralized algorithms for robotic control so that they could self-organize into certain patterns through local perception and communication between individuals to reduce the data transmission and communication cost between individuals and a central control station [46,53,54]. $H_{\mathrm{NDist}}$ could be used as an important measure for characterizing the overall configuration and for fine-tuning local interaction to achieve specific functions.

## 4. Methods

### 4.1. Boids Simulation

We adopted a simplified discrete-agent model with point-like boids whose dynamics are governed by the three rules: cohesion, separation, and alignment. Specifically, cohesion means each boid will tend to gather together and move to the center of its neighbors. Separation means if two boids get too close, they will tend to stay away from each other. The alignment refers to boids will move along the same direction as their neighbors. The size of the simulation box is 1000 × 1000 pixels. We employ a periodic boundary condition.

➢
*Algorithm*


The interaction between boids includes separation, alignment and cohesion force:(1)$$\left\{\begin{array}{c} F_{i,\mathrm{sep}}=\underset{j\in \varphi }{\sum }\frac{\left(r_{i}-r_{j}\right)}{{\left|r_{ij}\right|}^{2}}\\ F_{i,\mathrm{alg}}=\frac{\sum_{j\in \varphi }v_{j}}{n_{i}}-v_{i}\\ F_{i,\mathrm{coh}}=\frac{\sum_{j\in \varphi }r_{j}}{n_{i}}-r_{i} \end{array}\right.$$
where φ is the set containing all neighbors of ith boid.

To eliminate the influence of the absolute value of three kinds of interaction, we normalized these three parameters to obtain the total force:(2)$$F_{i}=\frac{F_{i,\mathrm{sep}}}{\left|F_{i,\mathrm{sep}}\right|}+\frac{F_{i,\mathrm{alg}}}{\left|F_{i,\mathrm{alg}}\right|}+\frac{F_{i,\mathrm{coh}}}{\left|F_{i,\mathrm{coh}}\right|}$$

Thus, we define the direction of movement at the moment t+Δt as:(3)$$\theta_{i}\left(t+\Delta t\right)=\arctan\left(\frac{v_{i}\left(t\right)+S\cdot F_{i}}{\left|v_{i}\left(t\right)+S\cdot F_{i}\right|}\right)$$
where S is the steering factor, which controls the relative weight between the influence from neighbors and the influence from the velocity of the boid. We tune S in the range from 0.001 to 0.1.

The velocity at moment t+Δt has contributions from $\hat{v}$ and $\hat{\theta }$,which represent the radial and angular components separately:(4)$$v_{i}\left(t+\Delta t\right)=V\hat{v}+\left(\theta_{i}+\Delta \theta \right)\hat{\theta }\;\;\Delta \theta \in \left[-\eta ,\eta \right]$$
where *V* is the constant speed, Δθ is a random variable drawn from a uniform distribution, and *η* is the noise factor in degree.

Then we could get the exact position of ith agent for the next moment by:(5)$$x_{i}\left(t+\Delta t\right)=x_{i}\left(t\right)+v_{i}\left(t+\Delta t\right)\Delta t$$

➢
*Definition of neighbors who has mutual interaction*


We defined neighbors who have influence on each other in two different ways. For metric distance standard in two-dimension, the set of neighbors φ includes all units within the circle whose radius is R center at particle i. The second way is defined by topology structure. Particularly, we perform Voronoi tessellation, and the neighbors are defined by a convex polygon.

### 4.2. Data Analysis


➢*Calculation of average velocity* $v_{a}$


The average velocity is the sum of the velocity of all boids:(6)$$v_{a}=\frac{1}{NV}\left|\sum_{i\in S}v_{i}\right|$$
where *S* is the set of all boids.


➢*Calculation of Entropy by neighbor distance* $H_{NDist}$


In calculating $H_{NDist}$, the neighbor always refers to topological neighbors. After determining neighbors by Voronoi tessellation, we obtain the distance between each agent and its neighboring agents. Then we count the neighbor distances of all agents to obtain the neighbor distance distribution of the overall configuration. We apply the formular of the Shannon entropy to calculate the entropy by neighbor distance
(7)$$H_{NDist}=-\sum_{i}p_{i}\log_{2}\left(p_{i}\right)$$
where $p_{i}=X_{i}/X$ is the probability of a neighbor distance that falls within a distance interval (a bin) labeled by index i, X is the total count of all neighbor distances of all agents, and $X_{i}$ is the count of the neighbor distances in bin i.

For the convenience of comparison, the normalized $H_{NDist}$ was introduced as:(8)$$H_{NDist,norm}=\frac{-\sum_{i}p_{i}\log_{2}\left(\frac{p_{i}}{w_{i}}\right)}{-log_{2}\left(\sum_{i}w_{i}\right)}$$
where $w_{i}$ is the bin size.

➢
*Particle tracking*


The image sequences were analyzed to investigate the structure and dynamics of the self-propelled active colloidal particles and robotic swarm. Particle tracking was performed by Python package Deeptrack (https://github.com/softmatterlab/DeepTrack2 (accessed on 23 July 2023)) [55] and Trackpy (https://soft-matter.github.io/trackpy/dev/index.html (accessed on 23 July 2023)) to identify the 2D location of each agent. The xy coordinates of the agents were used to calculate $H_{NDist}$ and characterize the structural order in the systems.

## Figures and Tables

**Figure 1 micromachines-14-01503-f001:**
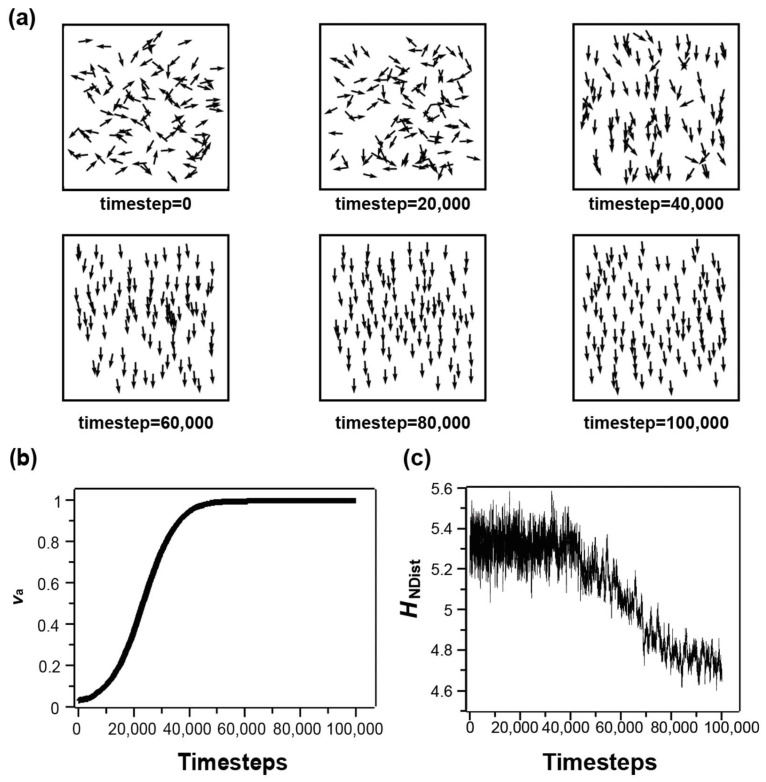
The analysis of one representative simulation run of the boid system (S = 0.1, topological neighbors), using entropy by neighbor distance $H_{\mathrm{NDist}}$ and magnitude of the averaged velocity $v_{a}$ (**a**) representative patterns at several key timesteps: arrows represent the velocity directions of boids. (**b**) the evolution of $v_{a}$, (**c**) the evolution of $H_{\mathrm{NDist}}$.

**Figure 2 micromachines-14-01503-f002:**
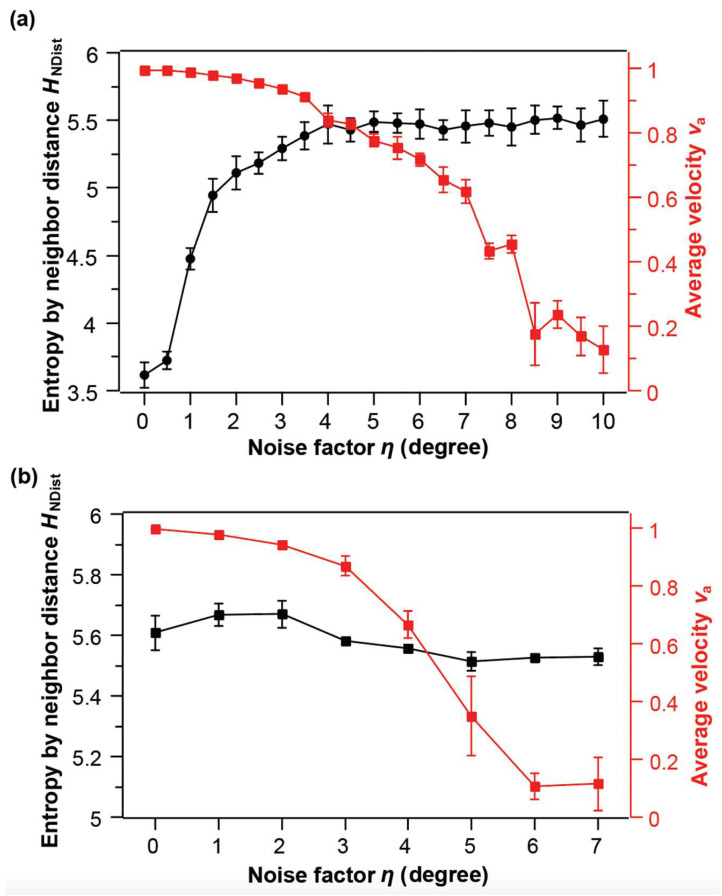
The entropy by neighbor distances $H_{\mathrm{NDist}}$ and the magnitude of the averaged velocities $v_{a}$ of final steady-state patterns in the simulated boids systems as a function of the noise factor η. The steering factor S is 0.001 in all cases. The error bars indicate the standard deviations obtained in at least three independent runs. (**a**) Neighbors are defined by topology. (**b**) Neighbors are defined by metric distance.

## Data Availability

We have uploaded our code to Github (https://github.com/wowoshu/boids (accessed on 23 July 2023)).

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
