# Peer review of "Entropy by Neighbor Distance as a New Measure for Characterizing Spatiotemporal Orders in Microscopic Collective Systems"

_micromachines, 2023, doi:10.3390/mi14081503_

Round 1

Reviewer 1 Report

The manuscript provides a method to characterize the ordering of the self-organized systems. The method calculates the Shannon entropy associated with the probability distribution of neighbor distances that has been reported in [Science 374 Advances 2022, 8, eabk0685]. In the present work, the authors apply the measure to three new systems. So, my major concern is the novelty of this work. The significance and contribution of the current work should be clearly demonstrated.

There are some minor points:

1. Figure caption should be carefully checked. Figure 1b is va and 1c is HNDist.

2. Page 4, line 137, the velocity vi is not defined clearly.

3. In Figure 3a, there is a significant difference between the values of HNDist at t =0 and 40 s when light is turned on, why? In Figure 3b, the second state is an amorphous aggregated state rather than a crystal-like state.

4. Page 11, line 290, the definition of average velocity is wrong! See [PRL, 1995, 75, 1226].

There are some English language issues, such as line 289 “Calculaation”.

Author Response

Please see the pdf version of the response letter.

Reviewer 2 Report

The authors demonstrate the characterization of collective patterns by the entropy of the neighbor distance distributions. The results are robust, and the content is well-organized. However, there are a few concerns that need to be addressed:

 (1)   The authors mentioned that “The comparison of Fig.1c and Fig.1d shows that the velocity alignment….”. However, no Fig. 1d is included in the manuscript.

(2)   Please check the captions of Fig. 1b and c.

(3)   The exact time points of “light on” and “light off” are not clearly labeled in Fig. 3.

(4)   It appears that the ?_NDist analysis does not provide significant information regarding pattern transitions in kilobot swarms (Fig. 4). Could the authors explain why the ?_NDist increases during pattern transitions?

(5)   The authors claimed that “Moreover, the final values of ?_NDist for the two shapes differ, which suggests ?_NDist may be used as an indicator for reconfigurable kilobot self-assemblies as well”. To support this claim, more data is needed.

(6)   Could the authors discuss the applicability of ?_NDist analysis in 3D collective systems?

(7)   I found some references related to collective systems (Nano-Micro Lett. 2023, 15, 141; ACS Nano 2020, 14, 406; iScience 2019, 19, 415).

The quality of the English writing is satisfactory.

Author Response

(The authors gave the same response as above.)

Reviewer 3 Report

Comments to the articles

The article written by Yelei Fu and et al. shows their new methods to depicts the spatiotemporal orders in microscopic collective systems by neighbor distance related entropy, which has been applied to three different scenarios. The work seems interesting, but the author didn’t spend time explaining the data and how to connect their simulation results with the experimental results well. Some of the results are just description of the plots and no hypotheses, conclusions are made. It needs to be further improved to show a better story rather than describe what you have done. Given all the considerations, I would suggest the article needs to be revised and refined before publishing in micromachines. And some remaining pointes need to be addressed as follows:

1.    First, it has been widely accepted that entropy can govern the collective behaviors of the building blocks into ordered structure, especially by Dr. Sharon Glotzer. They developed a series of new superstructures to show the university of this entropy driven self-assembly. In their work, by using entropy, they excluded all the interactions and used rigid body interactions to determine the entropy of the system. However, in your simulation, the separation and cohesion and alignment forces are involved, are these interactions can be also regarded as entropy? For the colloidal systems, from experimental observation, it is easy for us to know the average distance between colloidal particles will decrease if they start to aggregate, without HNDist, we can probably get similar conclusions. The key point here is how to determine the length scale of order or the transition points of the dispersed or gas-like state to aggregation or crystal-like state? So how to define the transition point in your measure?

2.    The arrangement of panel 1b and panel 1c or the caption in figure 1 is wrong, please double check your figures.

3.    The author mentioned that the average velocity and topological structures are two measures to characterize different aspects of the collective patterns. What are the aspects that we are caring about? Can you give some explanations of how average velocity and topological structure affects the collective patterns?

4.    For the firstly case, it seems that the simulation can work well in liquid crystal, can you apply your methods to determine the orders of liquid crystal and give us some ideas how this method can be used to address the issues that encountered in liquid crystals.

5.    Why does the noise factor (?) at 4% become the critical points for the transition of Va and HNDist? Does the noise factor change after changing the initial condition of your simulations? Is there any scientific reasons that 4% is the critical point?

6.    How did you get the HNDist and Va in Figure 2? I am assuming that you run the simulation like Figure 1b and 1c with different noise factor, and extract the HNDist and Va from your simulations, are these values from the final point in each simulation?  So it is better to indicate what data you are using.

7.    In Figure 3a, the light can induce the aggregation of the colloidal particles as shown by the decreased HNDist, which makes sense. However, when switching the light off, the HNDist increased a little bit and then decreased, and eventually increased; afterwards, the HNDist increased with light on, even larger than the initial states. How to explain that? In adtion, for the cluster formation, is it possible for you measure to distinguish individual particle from the clusters, especially at the very beginning stage that the cluster is small.

8.    In Figure 3b, the difference between HNDist of gas state like and crystal-like state small, around 15% difference. If given you an unknow system or dynamic system involve phase transition, by using HNDist, how can you tell whether it is a crystal-like state or gas state? The key point here is how to define the threshold of the transition from gas state to crystal-like state? By analyzing two already-know states and then comparing the HNDist seem not interesting, and people can easily tell based on the movie whether they are aggregated or not. Therefore, it is highly recommended to determine the boundary of two different states and give us the reason why the boundary is there.

9.    In figure 4, why HNDist increased with the formation of large self-assembly, which is counterintuitive. The author needs to explain this inverse trend. In addition, is the total number of the robot of k-shape and starfish-shape the same? Is it possible that the HNDist is number of robot dependent? If not, why the HNDist is shape dependent? The author needs to explain their idea in more detail.

Overall, this article showed an interesting measure to determine the spatiotemporal orders of the collective systems. The foundation of the article is interesting, but the explanation of results is lacking details. I think the quality of the article can be improved by addressing above mentioned concerns.

The language is decent and easy to follow.

Author Response

(The authors gave the same response as above.)

Round 2

Reviewer 1 Report

I recommend publication in current form.